# Exploring Data Distillation for efficient generation of Tabular Data

## Abstract

Tabular data generation methods have emerged to address growing concerns in the use of sensitive tabular data for training machine learning models. Many methods focus on creating high quality tabular data that can be used in place of the original dataset while retaining generalization performance on downstream tasks, and protecting sensitive data in an era where privacy is paramount. Despite their avid success, many of the methods face implacable challenges and obstacles to wide scale applications primarily due to the significant computational costs associated with data synthesis. In this paper, we propose a flexible data distillation pipeline as an alternative to conventional synthetic data generators that obtains competitive privacy metrics, while achieving significantly higher downstream performance at a fraction of the compute costs. In particular our methods has accelerated data synthesis by $5\times$ on average when compared to synthetic generators, while also achieving superior performance.

## 1 Introduction

Tabular data is one of the most ubiquitous mediums of data storage Shwartz-Ziv & Armon (2021) across fields such as medicine, physics, and financial institutions Sahakyan et al. (2021); Zabërgja et al. (2024), due to the flexible representation of high dimensional data in a structured form. In particular, tabular data contains a mixture of feature data types, including numerically continuous and categorically discrete data, often paired with a highly imbalanced class distribution Zhao et al. (2021b); Bennett (2001); Jolicoeur-Martineau et al. (2024). Further Tabular datasets often encode sensitive information hence sparking avid research in the field of tabular data synthesis Kotelnikov et al. (2023); Lee et al. (2023); Zhang et al. (2023). However, these synthetic generators often come at the expense of heavy training and deployment costs that scale with complexity of the dataset Zhang et al. (2023). In this paper we naturally pose the question: *Can we efficiently synthesize a compact dataset that retains downstream generalization performance ?*

In this work, we choose to frame this question as a *data condensation* problem, in which we aim to efficiently condense the original datasets into a small compact training dataset for downstream tasks, such as machine learning efficiency (MLE). In the image-classification domain, data condensation has gained significant popularity for efficiently reducing the computational costs of downstream training on image data upwards of 1 million training samples. These strategies, collectively referred to as data condensation, can be divided into core-set subset selection Guo et al. (2022); Rebuffi et al. (2017); Castro et al. (2018); Belouadah & Popescu (2020); Sener & Savarese (2018); Toneva et al. (2019) and dataset distillation techniques Wang et al. (2018); Zhao et al. (2021a); Zhao & Bilen (2023); Cazenavette et al. (2022); Sajedi et al. (2023); Wang et al. (2022); Zhao et al. (2023), both of which are implemented to drive down the computational costs of downstream training while maintaining generalization performance.

In this work, we present a novel alternative to synthetic data generators for tabular data through the use of data condensation techniques. In particular we leverage the use of an auto-encoder to project the inherent mixed-data types into a unified latent embedding where we can then efficiently condense the dataset into an increasingly private compact training dataset for downstream tasks. In Figure 1 we demonstrate the capacity for distilled data to server as effective synthetic data by comparing it's privacy preservation, accuracy on machine learning efficiency, and it's effectiveness at reducing computational costs at training and deployment time.

Figure 1: (Left) Compared to the real data from the original tabular, the privacy issue in the distilled tabular is much smaller. (Middle) Compared to generative-based methods, our proposed method shows a good trade-off between performance and privacy. (Right) Our approach signficantly improves the efficiency of training/data synthesis, and is also more efficient for full-scale deployment.

As one of the first works to introduce data condensation as an efficient method of synthetic data generation on real world tabular data, we begin by comparing the data distillation techniques with synthetic generators. In particular, in Section 4, we use a variety of metrics including computational efficiency, privacy through distance to closest record, and machine learning efficiency to ultimately justify the effective use of data distillation on tabular data. We then ablate the different method of data distillation against naive coreset selection techniques in Section 4.2 to demonstrate it's downstream performance benefits. We also evaluate our distilled data on a variety of tabular-based downstream models including gradient boosting networks. Finally, we illustrate the improved efficiency of parameter search using our distilled data as a proxy for fast and effective parameter selection.

## 2 PRELIMINARIES

In this section, we provide some background on the key challenges behind integrating data distillation techniques with tabular data, and then discuss our relevant design choices to facilitate this combination. Tabular data does present many additional challenges such as heterogeneity, column sparsity, data pre-processing, and even domain knowledge. However, in this paper, we aim to create an alternate means of protected data synthesis using data distillation, hence we focus particularly on the obstacles that hinder the straightforward application of data distillation; in particular *mixed data types* (heterogeneity) and *class imbalance* (sparsity).

**Mixed Data Types.** In tabular data, each column represents a particular feature that is distributed over the dataset rows, but not all features share the same data type Shwartz-Ziv & Armon (2021). Typically, a tabular dataset includes a mix of continuous numerical and discrete categorical features. In the former case, features span a continuous range, akin to pixel RGB values in images, while discrete features are categorical, similar to classification outputs. Data condensation pipelines, coreset and distillation techniques, typically target continuous data. However, applying these methods to a mixture of feature types presents challenges.

To address these challenges , we introduce the use of a variational autoencoder, following similar techniques from Zhang et al. (2023); Liu et al. (2022), to encode continuous and categorical features into a latent embedding space. Here, we can condense the dataset and use the decoder to transform our condensed embedding into a condensed tabular dataset. In order to fully confirm this hypothesis, we perform a test using one core-set and one distillation strategy on both the real and latent spaces (further details in the Appendix). As shown in Table 1, running distillation directly on the tabular data results in far

Table 1: Real vs. Latent dimension for information matching.

| Space | 5% | | 10% | |
|---|---|---|---|---|
| Real | 50.8 | | 60.2 | |
| Latent | 64.1 | ↑13.3 | 85.8 | ↑25.6 |

worse performance than condensing in the latent dimension. This implies that the latent dimension combines both continuous and categorical data into a unified, more comprehensive embedding space. Thus we use the latent dimension in our distillation pipeline methods.

**Class Imbalance.** Previous data-efficient training pipelines, often based on core-set selection or dataset distillation, rely on standardized image benchmarks for image classification tasks. These benchmarks typically maintain balanced classes across datasets. For instance, distillation strategies in image classification are typically evaluated based on the number of image samples per class (IPC) given equal class representation. However, tabular datasets frequently exhibit class imbalances.

For example, the *Adult* dataset has roughly three times as many positive classes as negative ones. Extending the concept of IPC to tabular data involves considering the number of rows/samples per class, however, this runs the risk of changing the underlying data distribution, which can negatively impact subsequent tasks, hence we propose a *distillation ratio per class* (DPC). To preserve the initial distribution, we use DPC, which retains the innate imbalance that exists in the underlying dataset. For example, a DPC of $1\%$ indicates a synthetic dataset with $1\%$ of the samples in each class. Throughout this paper, our goal is to create a framework that allows models to perform competitively with small DPCs.

## 3 METHODOLOGY

In this section, we introduce our novel tabular data condensation framework, which creates a platform for extending *any* dataset distillation method from the image domains into the tabular benchmarks. Formally, our framework condenses knowledge from a large-scale tabular training dataset $\mathcal{T} = \{(\boldsymbol{x}_i, y_i)\}_{i=1}^{|\mathcal{T}|}$, with $|\mathcal{T}|$ feature-label pairs, into a smaller, yet informative, dataset $\mathcal{S} = \{(\boldsymbol{s}_j, y_j)\}_{j=1}^{|\mathcal{S}|}$ that has comparable *machine learning efficiency* (i.e., downstream testing performance) with a model trained on the original dataset. We depict the overall pipeline in Figure 2.

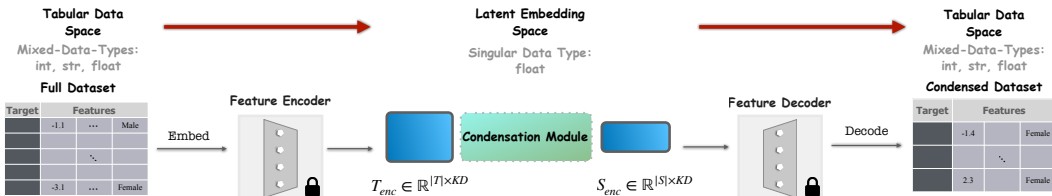

Figure 2: High-level depiction of our Flexible Tabular Condensation Framework. The full dataset is encoded into the latent embedding space, using a fixed pre-trained feature encoder. Data condensation is applied on the embedding prior to decoding the synthetic dataset.

Our primary goal in this work is to propose data distillation as a more efficient alternative approach to synthetic data generators. Following this approach, we begin by describing our overall framework, as well as provide details on three specific distillation algorithms extended from visual domain. These methods provide foundation of many current state of the art works in the domain, hence, applying them to tabular data opens the door for future iterative work in the field.

### 3.1 FLEXIBLE TABULAR CONDENSATION FRAMEWORK

As the first method to expand dataset condensation to "real-world" tabular data, we establish a unified framework for applying different distillation strategies in the context of tabular data. In particular, our framework contains two main components: an *autoencoder pipeline* and a *condensation pipeline*. Figure 2 provides a high-level view of our framework.

### 3.2 AUTO-ENCODER PIPELINE

Following previous tabular works Zhang et al. (2023); Liu et al. (2022); Xu et al. (2019); Gorishniy et al. (2021), we employ a dataset-specific pre-trained VAE-Transformer for encoding the full dataset into a latent embedding dimension and decoding the synthetic dataset into the tabular data space (see Figure 2). Since the synthetic dataset is designed to "match" the informative content of the full dataset, we find it sufficient to freeze the weights of the VAE after pre-training. Our Autor-encoder pipeline is composed of 2 components adapted from Zhang et al. (2023): (1) Tokenization, (2) Embedding Map.

**Tokenizer** Following the successful works of TabSyn Zhang et al. (2023), we use a tokenizer to convert the mixture of numerical and categorical columns into a unified representation. Formally, a single row in the tokenized dataset can be represented as a collection of tokens Zhang et al. (2023)

$$Sample \leftarrow [t_0, \cdots t_i \cdots t_N] \in \mathbb{R}^{N \times D} \tag{1}$$

Importantly we identify that there is a reversible tokenizer to transform the data back into the mixture of continuous and discrete samples enabling downstream training.

**Embedding Map** Given a tokenized input we leverage a VAE encoder to project our samples into an embedding space. Following the works of TabSyn Zhang et al. (2023), the encode generates a mean and log variance which can be transformed into a latent embedding with reparametrization ($Z = \mu + \sigma * \alpha, \alpha \in \mathcal{N}(0,1)$).

### 3.3 PIPELINE

Formally, given the full tabular dataset $\mathcal{T}$, we derive an encoding as $\mathcal{T}_{enc} = \texttt{Encoder}(\mathcal{T})$, where $\mathcal{T}_{enc} \in \mathbb{R}^{|\mathcal{T}| \times K \times D}$. Here, $K$ and $D$ represent the latent token and embedding dimensions, respectively, for a transformer-based encoder ($K$ equals the number of features in the dataset, and $D = 4$ following Zhang et al. (2023); refer to the appendix). This process applied both tokenization and the embedding map described previously. Applying the selected distillation strategy on the features, we obtain a small set of synthetic embeddings ($\mathcal{S}_{enc} \in \mathbb{R}^{|\mathcal{S}| \times K \times D}$), after which we use the pre-trained decoder to reverse the process. Formally, we obtain the real tabular features of the synthetic dataset as $\mathcal{S} = \texttt{Decoder}(\mathcal{S}_{enc}) \rightarrow \{(\boldsymbol{s}_j, y_j)\}_{j=1}^{|\mathcal{S}|}$. Note, the labels are not used explicitly in the encoding/decoding process, as condensation strategies are applied per class; hence, labels are appended after decoding to yield the synthetic dataset.

### 3.4 DATASET DISTILLATION STRATEGIES

Throughout this study, we incorporate three primary dataset distillation strategies: Distribution Matching, Attention Matching, and Gradient Matching. Here, we delve into the intricacies of each method and the associated adaptations made to integrate their distillation strategies into our flexible pipeline. Visually, we illustrate the creation of our synthetic set $\mathcal{S}$ using distillation

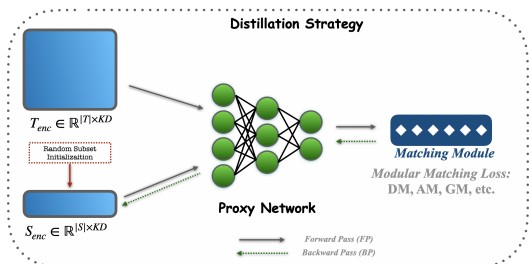

Figure 3: Dataset Distillation Module

in Figure 4. While Trajectory Matching is another method prevalent in the image domain, due to its significantly high computational requirements, we find that feature matching and gradient matching suffice. To provide context, trajectory matching costs nearly $5\times$ the computational runtime of feature matching and $2.5\times$ the cost of gradient matching Sajedi et al. (2023). We leave this extension to future work but show that our chosen distillation strategies excel in tabular distillation. All distillation methods are applied with default hyperparameters (see more details in the Appendix).

**Distribtuion Matching (DM).** DM, first proposed in Zhao & Bilen (2023), is a computationally efficient method for dataset distillation evading bi-level optimization by employing an untrained proxy network to match the embedding features of real and synthetic images. DM follows class-by-class aggregation to account for a class's mean distribution, by leveraging a ConvNet network Gidaris & Komodakis (2018) to capture spatially encoded features in images. Given a feature extractor $\psi(\cdot)$ from the class of randomly initialized networks, and the images from the real $\mathcal{T}_c$ and synthetic $\mathcal{S}_c$ dataset for class $c$, the distribution matching loss can be defined as:

$$\mathcal{L}_{DM} = \sum_c \left\| \frac{1}{|\mathcal{T}_c|} \sum_{\boldsymbol{x} \in \mathcal{T}_c} \psi(\boldsymbol{x}) - \frac{1}{|\mathcal{S}_c|} \sum_{\boldsymbol{s} \in \mathcal{S}_c} \psi(\boldsymbol{s}) \right\|^2. \tag{2}$$

**Adapting DM to Tabular Data** required changes to both the proxy network and the feature matching loss. Instead of using $\psi(\cdot)$ to encode raw pixels into an embedding space, we work directly with the latent embeddings of our tabular data as input. We sample the latent vectors from the real dataset belonging to a particular class of interest, as well as our synthetically initialized latent vectors for the same class, and feed them through a randomly initialized proxy network to perform class-by-class matching. Essentially, we replace the image-based $\mathcal{T}_c$ and $\mathcal{S}_c$ with our encoded latent vectors $\mathcal{T}_{enc}$ and $\mathcal{S}_{enc}$ as defined in Section 3.2, allowing us to learn the synthetic set. We also replace the typical ConvNet with a simple 3-layer MLP to handle our token-based parameterized embedding dimension, as spatial extraction is not relevant in tabular embedding. After completing the distribution matching process, we return $\mathcal{S}_{enc} \in \mathbb{R}^{|\mathcal{S}| \times K \times D}$, which can be decoded into the synthetic tabular dataset.

**Attention Matching (AM).** AM was first proposed in Sajedi et al. (2023) to enhance distribution matching by utilizing attentive scores from intermediate feature maps. Two methods of attention matching have been used in image distillation: spatial attention Sajedi et al. (2023) and channel-wise attention Khaki et al. (2024). Formally, given a class of interest $c$ and a particular layer $l$ in the

randomly initialized proxy network $\psi(\cdot)$, the attention matching loss can be defined as:

$$\mathcal{L}_{AM} = \sum_c \sum_l \left\| \frac{1}{|\mathcal{T}_c|} \sum_{\boldsymbol{x} \in \mathcal{T}_c} \left[ \frac{\boldsymbol{z}_l(\boldsymbol{x})}{\|\boldsymbol{z}_l(\boldsymbol{x})\|_2} \right] - \frac{1}{|\mathcal{S}_c|} \sum_{\boldsymbol{s} \in \mathcal{S}_c} \left[ \frac{\boldsymbol{z}_l(\boldsymbol{s})}{\|\boldsymbol{z}_l(\boldsymbol{s})\|_2} \right] \right\|^2, \tag{3}$$

where $\boldsymbol{z}_l(\boldsymbol{x})$ denotes the operator that obtains the attention map of input $\boldsymbol{x}$ at layer $l$ in the proxy network. For normalized channel-wise attention, an input sample is represented as $\boldsymbol{x} \in \mathbb{R}^{ch \times d}$, where $ch$ is the channel dimension and $d$ represents the vectorized spatial dimensions Khaki et al. (2024).

**Adapting AM to Tabular Data.** Once again, we, adopt an MLP as our proxy network, hence altering the intermediate dimensions used for attention matching. Our VAE encodes an embedding space where a single sample is parameterized by $\boldsymbol{x}_0 \in \mathbb{R}^{K \times D}$, with $K$ representing tokenized dimensions. By defining spatial and channel-wise attention for combining information from different filters, we can apply similar techniques to transformer-based embeddings. Given our latent embedding space's dimensional arrangement, we can exchange "channels" for "tokens," where channels capture different spatial localizations in images and tokens represent different textual localizations in the tabular dataset. We compute token-wise attention, using the remaining embedding dimension to match real and synthetic datasets. We redefine $\mathcal{L}_{\text{ATOM}}$ from Khaki et al. (2024) to use attention along the tokens of the intermediate layers. Specifically, we compute $\mathcal{L}_{AM}$ using our latent vector, with attention to the tokens (instead of channels) and the textual embedding dimension $d$ (in place of the spatial embedding), denoted as *token-based-attention*. This approach aligns the embeddings of each feature (column) between real and synthetic datasets, providing a meaningful representation for each attribute.

**Gradient Matching (GM).** GM Zhao et al. (2021a) uses bi-level optimization to guide the learning process with a network trained on synthetic data, typically favoring samples with high gradients. Despite the inherent bias based on gradient magnitude, this core-matching principle has been adopted in various new distillation methods, including DSA Zhao et al. (2021a), Sequential Matching Du et al. (2024), and Contrastive Signals Lee et al. (2022). Given the weight parameters $\theta$ of a trained proxy network $\psi(\cdot)$ and an objective function $l$ (cross-entropy for classification), we compute the gradients with respect to the real and synthetic data as $\nabla_\theta l(\theta, \mathcal{T})$ and $\nabla_\theta l(\theta, \mathcal{S})$, respectively. Formally, the gradient matching loss can be defined as:

$$\mathcal{L}_{GM} = 1 - \frac{\nabla_\theta \ell(\theta, \mathcal{S}) \cdot \nabla_\theta \ell(\theta, \mathcal{T})}{\|\nabla_\theta \ell(\theta, \mathcal{S})\| \, \|\nabla_\theta \ell(\theta, \mathcal{T})\|}. \tag{4}$$

**Adapting GM to Tabular Data.** The typical trained proxy network (ConvNet) is not well-suited for handling the mixture of tabular data types. Similar to Distribution and Attention Matching, we replace the proxy network with a simple multi-layer perceptron (MLP). Additionally, we add a classification head to enable bi-level optimization, supervised by the latent embeddings and corresponding label vectors. Using the trained proxy network, we determine the gradients using cross-entropy (objective loss) with samples of the real and synthetic latent vectors. Backpropagation is performed over the model onto the synthetic latent vectors. Incorporating an internal training loop for the MLP using embeddings and labels is sufficient for extending gradient matching-based distillation to tabular data.

## 4 Experimental

**Datasets.** To demonstrate the versatility of the proposed framework, we deploy 7 real-world tabular datasets with a mix of categorical and continuous attributes. Following previous works Zhang et al. (2023), we include a selection of four binary datasets from the UCI repository Asuncion & Newman (2007): Adult, Shoppers, Default, Magic. Notably, we also evaluate methods on three multi-label datasets: Covertype, Wine and Fourier. The overall data statistics include sample counts up to 500K, with over 50 attributes and up to 7 categorical class labels. Further details of the individual data statistics can be found in the appendix.

**Baseline Comparisons.** We begin our experimental section by comparing with current state of the art synthetic data generators including TabSyn Zhang et al. (2023),STaSY Kim et al. (2022b), TabDDPM Kotelnikov et al. (2023), and GReAT Borisov et al. (2022b). We further evaluate different data distillation techniques implemented in our framework: DM Zhao & Bilen (2023), AM and DC . We additionally ablate the usage of various core-set selection baselines Random, Least Confidence and K-Center Guo et al. (2022).

**Evaluation Network.** Following previous tabular works Liu et al. (2022); Zhang et al. (2023), we employ a simple multi-layer perception for the task of evaluating machine learning efficiency on all

| Method | Avg. Synthesis Time | Avg. # of Trainable Prams | Avg. Deployment Time | DCR | Adult | | Wine | | Default | |
|---|---|---|---|---|---|---|---|---|---|---|
| | | | | | 5% | 10% | 5% | 10% | 5% | 10% |
| *Original Dataset* | | | | | | | | | | |
| - | - | - | 100% | 85.6 | | 95.6 | | 72.6 | | |
| *Generative Models* | | | | | | | | | | |
| TabSyn | 4353 | ∼ 10M | 12.0 | 67.8 | 61.7 | 63.0 | 78.8 | 83.7 | 52.1 | 59.2 |
| STaSY | 3367 | ∼ 10M | 36.5 | 62.2 | 61.4 | 62.2 | 61.1 | 73.3 | 60.0 | 62.5 |
| GReAT | 18024 | ∼ 10M | 755 | 68.7 | 61.8 | 63.1 | 62.4 | 75.8 | 61.5 | 64.3 |
| TabDDPM | 2150 | ∼ 10M | 145.2 | 69.3 | 62.5 | 64.3 | 80.2 | 85.3 | 67.3 | 67.9 |
| *Distillation Methods* | | | | | | | | | | |
| DM (ours) | 1102 | 29.1K | 2.5 | 65.1 | 64.1 | 85.8 | 90.5 | 92.6 | 71.2 | 71.2 |
| AM (ours) | 1203 | 29.1K | 2.5 | 65.5 | 84.2 | 86.8 | 93.2 | 98.5 | 71.0 | 71.1 |
| GM (ours) | 1422 | 130.8K | 2.5 | 67.2 | 86.0 | 86.8 | 90.3 | 93.5 | 71.1 | 71.2 |
| Improvement | ↑5.5× | ↑15.5× | ↑100× | ↑1.1 | ↑13.5 | ↑22.3 | ↑13.0 | ↑8.2 | ↑3.9 | ↑3.3 |

Table 2: **Results on Tabular Dataset Benchmark Suite.** We compare the computational costs, privacy, and accuracy of traditional data generators with the proposed distillation framework on a variety of datasets. In all cases we show that distillation approaches can significantly reduce computational costs, while maintaining privacy with superior performance. For fairness, all methods use the same number of samples for downstream machine learning efficiency. Privacy metrics (DCR) are compute on the Adult dataset.

real and synthetic datasets. This network consists of 2 hidden layers with embedding dimensions of 100, trained for a maximum of 100 iterations at a learning rate of 0.001. This follows the conventional MLP learning settings established in Zhang et al. (2023). Additionally for select tasks, we include results on XGBoost, AdaBoost, Random Forest and Decisions Trees. Further details on these configurations can be found in the appendix.

**Performance Metrics.** Following TabSynZhang et al. (2023), we report classification performance using the area under the receiver operating characteristic (AUROC). For compute metrics we define the following terminology.

1. *Synthesis Time*: The time in (s) required to train a synthetic generator or distill a dataset

2. *Deployment Time*: The time in (s) taken to sample a synthetic generator and train the downstream model on the sampled dataset.

3. *# of Trainable Params*: The number of parameters in the generator or the number of parameters optimized in the distilled dataset.

All latency measurements are conducted on one NVIDIA A6000 GPU with a 20%-cycle warm-up. Finally, our privacy metrics are measured with distance to closest record (DCR), following the previous works Zhang et al. (2023). Vanilla DCR calculates the distribution of distances between two datasets. A value closer to 0 indicates high similarity between them. We present the DCR probability (%) between the original dataset and a partitioned hold-out set, following the experimental setup outlined in Zhang et al. (2023) (further details in the appendix). A DCR probability of 50% suggests that the synthetic dataset is equally distant from both the hold-out and training datasets, indicating a *low privacy risk*. Probabilities nearing 100% signify high overlap with the original training dataset, while 0% indicates high overlap with the holdout set.

### 4.1 EFFICIENT SYNTHESIS AND DEPLOYMENT

In this section, we compare the performance of SOTA synthetic data generators for tabular data with our proposed framework for data distillation. In recent years, there has been an avid surge in the development of synthetic tabular generators, aiming to produce similar data to the original dataset. Although there are many applications for synthetic generators, we show in Table **??** that these generators do not condense the information, and thus the produced synthetic data from generators performs synonymous with random subset selection, indicating the inability to deliver highly compact data for efficient synthesis or deployment. We compare the performance on machine learning efficiency using constant sample counts (DPC) for a mixture of binary and multi-class datasets. We additionally report several compute metrics and average privacy analysis as DCR. In Table **??**, we show that distillation approaches save synthesis time by almost $5.5\times$ with $15.5\times$ less trainable parameters. Further our distilled datasets maintain the privacy standard of current SOTA, with far superior performance in machine learning efficiency at reduced deployment costs.

## 4.2 Comparing methods of Data Condensation

Having established data distillation as a powerful method of accelerating tabular data synthetis while retaining performance and privacy metrics, we further ablate the type of distillation strategy used. In this section we compare our supported distilation strategies with naiive core-set selection on wider range of distillation ratios and datasets. Core-set selection strategies, albeit a promising data condensation avenue, are inhertently efficient subset selection techniques, thus they may retain the generlization performance, but will leak 100% of selected records, hence cannot be used for privacy cases. Nonetheless, they form a good baseline to illustrate the improvements that data distillation garners in both privacy and performance.

**Machine Learning Efficiency**

One of the primary goals for dataset distillation is to generate a small compact dataset that can be used to accelerate downstream training tasks. Our adoption of data distillation has been primarily focused on maintaining competitive privacy metrics while outperforming synthetic data generators with lower synthesis and and deployment times. In this section, we ablate the use of distillation strategies against naive core-set subset selection methods. Machine learning efficiency measures the utility of a dataset by comparing final performance of training on the synthetic versus original dataset. Our results in Table 3 show significant improvements across a variety of single and multi-class datasets with a minimal relative performance drop of $\leq 2\%$ from the full dataset using a 10% compression ratio.

| Method | Adult | | | Shoppers | | | Covertype | | | Magic | | | Fourier | | |
|---|---|---|---|---|---|---|---|---|---|---|---|---|---|---|---|
| | 0.05% | 5% | 10% | 0.05% | 5% | 10% | 0.05% | 5% | 10% | 0.05% | 5% | 10% | 0.05% | 5% | 10% |
| Core-Set Baselines | | | | | | | | | | | | | | | |
| Random | 51.0 | 61.5 | 62.1 | 67.3 | 75.7 | 79.1 | 81.4 | 90.7 | 91.4 | 66.2 | 87.7 | 88.9 | 84.5 | 94.7 | 96.6 |
| Least Confidence | 53.8 | 61.8 | 62.7 | 37.0 | 68.4 | 73.1 | 69.3 | 84.5 | 87.8 | 60.9 | 73.2 | 76.9 | 75.1 | 90.4 | 92.9 |
| K-Center | 50.7 | 62.5 | 63.7 | 40.2 | 57.4 | 81.3 | 77.8 | 90.0 | 91.8 | 69.7 | 81.5 | 88.5 | 84.0 | 95.8 | 97.1 |
| Distillation Methods (Ours) | | | | | | | | | | | | | | | |
| DM | 61.1 | 64.1 | 85.8 | 70.7 | 84.0 | 85.6 | 85.2 | 90.8 | 91.8 | 78.9 | 89.2 | 89.3 | 96.8 | 97.2 | 97.5 |
| AM | 67.6 | 84.2 | 86.8 | 70.3 | 82.5 | 86.0 | 86.5 | 90.8 | 91.7 | 80.9 | 88.6 | 89.1 | 96.4 | 97.3 | 97.6 |
| GM | 61.7 | 86.0 | 86.8 | 72.8 | 81.7 | 85.2 | 83.2 | 90.9 | 91.7 | 77.0 | 88.4 | 89.1 | 90.5 | 95.4 | 97.1 |
| **Improvement** | ↑13.8% | ↑23.5% | ↑23.1% | ↑5.5% | ↑8.3% | ↑4.7% | ↑5.1% | ↑0.2% | 0.0% | ↑11.2% | ↑1.5% | ↑0.2% | ↑12.3% | ↑1.6% | ↑0.5% |
| Original Dataset | | | | | | | | | | | | | | | |
| Full Dataset | | 85.6 | | | 90.3 | | | 92.2 | | | 92.0 | | | 98.2 | |

Table 3: **Results on Tabular Dataset Benchmark Suite.** We compare Distillation and coreset strategies on a variety of binary and multi-class datasets. Ultimately we show that distillation works much better with significant improvements from core-set strategies while retaining privacy metrics.

**Visualizing Data Selection Strategies.** In this section, we use t-stochastic neighborhood estimation (tSNE) to project our synthetic datasets (derived through distillation at 0.05%) over top of the original full dataset on the *Adult* dataset in Figure 4. We additionally include core-set sub-set selections to illustrate how direct subsets would be visualized in this representation. Ultimately, we show that the three distillation strategies can effectively capture strategic points in the distribution. Additionally, we visually observe that the distillation strategies have better coverage of the full embedding space, likely due to an iterative learning process, meanwhile, the core-set or subset methods exhibit a more geometrically clustered shape.

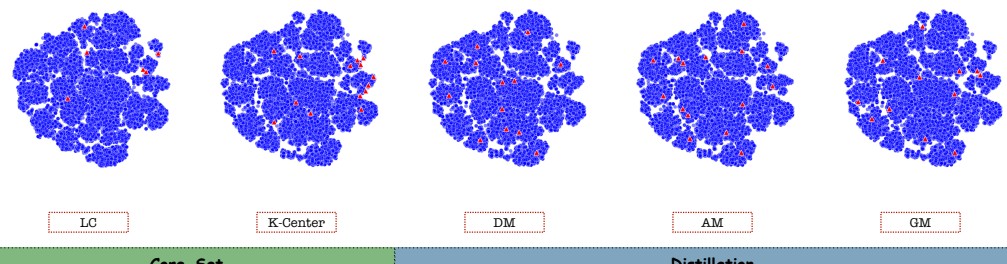

Figure 4: tSNE of our condensed datasets projected on the full dataset for *Adult* at a condensation ratio (DPC) of 0.05%. Blue circles (○) represent the underlying full-dataset, whereas red triangles (△) represent the synthetic dataset.

**Cross-Architecture Generalization**

Exploring the effect of different downstream architectures on the synthetic dataset is crucial for demonstrating generalization. In our condensation pipeline, only the distillation strategies use an

| Method | MLP | XGBoost | Random Forest | Descision Tree | AdaBoost | Average |
|---|---|---|---|---|---|---|
| Adult | | | | | | |
| Distribution Matching | 85.8 | 88.7 | 89.3 | 80.4 | 86.6 | 86.2 |
| Attention Matching | 86.8 | 89.2 | 89.7 | 82.1 | 87.8 | 87.1 |
| Gradient Matching | 86.8 | 90.3 | 90.3 | 84.8 | 89.2 | 88.3 |
| Full Dataset | 85.6 | 92.1 | 91.3 | 86.0 | 91.8 | 89.3 |
| Shoppers | | | | | | |
| Distribution Matching | 85.6 | 89.7 | 90.2 | 83.3 | 82.8 | 86.3 |
| Attention Matching | 86.0 | 90.0 | 90.3 | 83.8 | 81.5 | 86.3 |
| Gradient Matching | 85.2 | 89.9 | 89.9 | 82.6 | 81.7 | 85.9 |
| Full Dataset | 90.3 | 89.8 | 92.1 | 86.0 | 90.6 | 89.8 |
| Default | | | | | | |
| Distribution Matching | 71.2 | 73.8 | 75.9 | 67.7 | 73.2 | 72.4 |
| Attention Matching | 71.1 | 74.2 | 76.0 | 67.1 | 73.4 | 72.4 |
| Gradient Matching | 71.2 | 74.2 | 75.7 | 68.1 | 73.5 | 72.5 |
| Full Dataset | 72.6 | 75.9 | 77.2 | 67.6 | 76.7 | 74.0 |
| Magic | | | | | | |
| Distribution Matching | 89.3 | 91.8 | 91.9 | 84.1 | 89.4 | 89.3 |
| Attention Matching | 89.1 | 91.9 | 91.8 | 85.2 | 89.3 | 89.5 |
| Gradient Matching | 89.1 | 92.0 | 92.1 | 85.5 | 89.6 | 89.7 |
| Full Dataset | 92.0 | 94.6 | 94.3 | 89.7 | 91.6 | 92.4 |

Table 4: **Results on Tabular Dataset Benchmark Suite.** We compare our the machine learning efficiency using our distilled datasets on a variety of downstream models, and show maintained performance beyond the MLP architecture.

intermediate proxy network, as discussed in Section 3.4. Therefore, it is important to examine the impact of different architectures on the distilled data, a common practice in conventional dataset distillation Sajedi et al. (2023); Khaki et al. (2024); Zhao & Bilen (2023); Zhao et al. (2021a); Wang et al. (2022). In this section, we investigate the downstream effects of common tabular techniques, including XGBoost Chen & Guestrin (2016), Random Forest Breiman (2001), Decision Trees Hastie et al. (2009), and the AdaBoost Classifier Freund & Schapire (1997). In Table 3, we show that our distilled synthetic datasets maintain strong performance across various models. This signifies that our learned datasets incorporate globally important features in non-architecturally specific datasets. DM, AM, and GM naturally exhibit positive architecture transferability in the image domain Zhao & Bilen (2023); Sajedi et al. (2023); Khaki et al. (2024); Zhao et al. (2021a), hence we empirically confirm that the property of generalizability is retained in our framework.

**Computational Costs.** In this section, we explore the downstream effects of our condensed datasets. In this experiment we run a parameter search on XGBoost using the full dataset and the distilled dataset. In Figure 5 we compare the time distribution of parameter search when using the full dataset versus distillation. In the former, the entire search time is allocated to finding the ideal parameters, meanwhile in the latter we first train a VAE to encode the data, then distill, and finally run the search with our distilled proxy set. We can see in end-to-end time on the right, that our distillation method enables signficantly faster searching with very minimal perforamnce degradation.

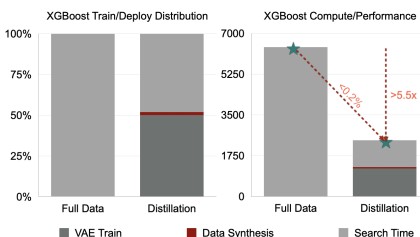

Figure 5: XGBoost Parameter Search

## 5 RELATED WORKS

Recent works in Tabular data have been mainly focused on the generation of new synthetic data that improves privacy while maintaining strong performance on several metrics such as data diversity, fidelity, and most commonly dowsntream machine learning efficiency (MLE). Despite their avid success in exceeding these metrics, most existing methods suffer from high training and generative inference costs often hindering real-world applications by imposing significant computational chal­lenges. In this paper, we specifically target MLE from the lens of data condensation. We present an

alternative approach that both improves the efficiency of creating synthetic data, as well as downstream MLE training with near lossless performance degradation. Unlike previous methods, our platform enables the acceleration of **both** data synthesis and model training.

**Tabular Data.** Tabular datasets have gained significant attention in recent years for applications in medicine and finance Kadra et al. (2021). They have been described as the "last unconquered castle" for deep learning due to the superior performance of traditional machine learning methods over modern architectures Kadra et al. (2021); Borisov et al. (2022a). Tabular data inherently contains a mixture of feature types, including continuous numerical and sparse categorical Borisov et al. (2022a). Numerous works focus on synthetic generators to produce new datasets Zhang et al. (2023); Liu et al. (2022); Zhao et al. (2021b); Borisov et al. (2022b); Lee et al. (2023); Kotelnikov et al. (2023); Kim et al. (2022b). In particular, TabSyn Zhang et al. (2023) uses score-based diffusion to acclerate the generation of robust tabular data. Likewise TabDDPM Kotelnikov et al. (2023) finds diffusion based probabilistic model to be an effective medium for generating data with good machine learning efficiency and moderate privacy metrics. In this paper we proposed data distillation as a new method of data synthesis which improves train/data synthesis time, as well as deployment time while maintaining superior performance and competitive privacy metrics.

**Dataset Distillation** Dataset distillation has emerged in computer vision for generating small, synthetic, and informative datasets that enable efficient training of downstream tasks with significantly reduced computational resources. Recently, it has accelerated applications in continual learning Chen et al. (2024); Sajedi et al. (2023); Zhao et al. (2021a); Gu et al. (2024); Yang et al. (2024), neural architecture search Ho & Ermon (2016); Such et al. (2020), privacy protection Dong et al. (2022); Chen et al. (2022); Chung et al. (2024); Loo et al. (2024), and federated learning Jia et al. (2023); Xiong et al. (2023); Liu et al. (2023a;b). Common methods include gradient matching Zhao et al. (2021a); Zhao & Bilen (2021); Lee et al. (2022); Kim et al. (2022a); Du et al. (2024), which aligns network gradients from real and synthetic datasets; feature and attention matching strategies Zhao & Bilen (2023); Wang et al. (2022); Sajedi et al. (2023); Zhao et al. (2023); Zhang et al. (2024), which align feature distributions between real and synthetic data in diverse latent spaces; and trajectory matching Cazenavette et al. (2022); Du et al. (2023); Cui et al. (2023); Guo et al. (2024), which minimizes differences in model training trajectories between original and synthetic samples. Most of these methods have been deployed exclusively on visual tasks, with some extending to adjacent modalities, including vision-language Wu et al. (2023), graph Jin et al. (2021), and a simulated toy-table dataset Medvedev & DâĂŹyakonov (2021). In this paper, we present the first framework to extend dataset distillation to real-world tabular datasets, incorporating distribution matching (DM) Zhao & Bilen (2023), attention matching (AM) Sajedi et al. (2023); Khaki et al. (2024), and gradient matching (GM) Zhao et al. (2021a).

# 6 LIMITATIONS

Our tabular data condensation pipeline, as the first effort in this domain, to propose distillation as alternative to data generators, involves empirically supported design choices, such as the use of an auto-encoder to handle mixed data types and a condensation ratio to address imbalanced datasets. Despite our significant performance, our method overlooks the varying feature correlations present in tabular data. Some features exhibit non-uniform effects within a dataset, with certain features being more sensitive to slight changes in numerical or categorical values. Although we encode both continuous and categorical values into an embedding space, we do not explicitly consider the differing "importance" of each feature.

# 7 CONCLUSION AND FUTURE WORK

We have developed a data distillation framework for efficient synthesis and deployment of real-world tabular datasets. Our framework addresses the challenges of mixed data types and class imbalances by introducing an auto-encoding pipeline and a class-specific condensation ratio. We demonstrate versatility by supporting three distillation strategies, allowing users to balance compression speed with performance. Additionally, we compare with state of the art data generators in terms of computational costs in both train and deployment time, as well as privacy and downstream performance in machine learning efficiency. We conducted extensive experiments on various real-world datasets, architectures and parameters demonstrating the transferability of our condensed datasets. Our method better condenses information into small synthetic datasets as opposed to conventional generators. In the future, we aim to address feature sensitivity by expanding distillation along the feature dimension.

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

# A  APPENDIX

This appendix includes supplementary information to help reinforce content that may have been discussed briefly in the main paper. In particular, Section A.1 provides more context on our design choices, and Section A.2 provides more details on the experimental configurations, datasets, metrics, and a brief discussion of our source code. Please also find our source code attached.

## A.1  DETAILS ON DESIGN CHOICES

### A.1.1  DETAILS ON ADDRESSING THE MIXED DATA TYPES

In the main manuscript, we describe the comparisons between distilling in latent and real spaces. Our results in Table 1 confirm that distillation and core-set selection should be performed in the latent space for the best results. Below, we describe how distillation and selection work in the `Real` and `Latent` spaces, respectively.

**Real Space.** In real space, selection mechanisms are modified to operate on a mixture of continuous numerical and discrete categorical features. For example, geometric methods such as K-Center use a distance metric computed as the norm separation between samples. However, in the case of categorical data, this is no longer generalized. Hence, we employ distance matching for categorical attributes, where all non-matching categories are assigned a distance of one while matching receives a distance of 0. Similarly, for distillation, we currently do not have a defined method for learning discrete categorical data; hence, in this case, we fix the categorical features and simply learn the continuous features directly in the real space; hence, parameter sensitivity is not accounted for with embedding. We do show some consistent performance; however, it only uses a subset of the learnable features and could cause contradictions in the underlying correlation between features as only some of them are learned. For these reasons, we recommend against distilling directly in real space.

**Latent Space.** Throughout this work, we use the latent space for distillation by converting the mixture of numerically continuous and categorically discrete data into a unified continuous embedding (the latent space). We describe this process using the auto-encoder in our main paper, and it is a method of choice for all distillation and selection experiments.

### A.1.2  DETAILS ON THE AUTOENCODER PIPELINE

For this work, we adopt the described auto-encoder design from TabSyn Zhang et al. (2023). We begin by tokenizing each column in the original tabular data into a d-dimensional vector. For the purpose of this work, we stick to a default dimension of $d = 4$. We can now express each data sample as a matrix defined by the number of features and the embedding dimension $d$. Given that we use a transformer encoder, the process of converting numerical and categorical features into their embeddings is learnable. Explicitly, numerical features are embedded using a learnable linear transformation; meanwhile, categorical features are pre-processed with one-hot encoding and embedded using a lookup table. Given a unified continuous embedding to represent the numerical and categorical features, a typical VAE is employed. The VAE-encoder obtains the mean and log variance of the latent, where the embedding is captured through reparameterization of the latent space. At this point, we can apply our selection and distillation strategies directly to this latent embedding. Following the completion of our methodology, we can pass the condensed embeddings into the VAE-Decoder to re-obtain a tokenized matrix. From this tokenized matrix, we can apply the symmetrically defined de-tokenizer (decoding transformer) to re-convert tokens into the real tabular space. Hence, from a high-level overview, tabular data is tokenized using a transformer, embedded through an encoder, condensed in the latent space, followed by decoding into tokens and detokenizing back into the real tabular dataset. We note that our method leverages the exact VAE-Transformer design from TabSyn Zhang et al. (2023), hence further details are included in their work.

## A.2  EXPERIMENTAL IMPLEMENTATIONS

### A.2.1  DETAILS ON DISTILLATION STRATEGIES

For the most part, we follow standard hyperparameters for distribution matching, attention matching, and gradient matching. Some slight modifications were made to the number of inner-loop training

epochs and the scaling of the loss propagation; however, for reproducibility, we include all these as default settings in our attached code. The

### A.2.2 DETAILS ON EVALUATION NETWORK

Given that our task of data distillation focuses on retaining generalization performance while reducing the size of the dataset, we found it most intuitive to work with an evaluation network that has many learnable parameters as opposed to a more rule- or heuristic-based method. Given the success of MLPs in the tabular domain, we adopted the MLP architecture for our Machine Learning Efficiency task. Conventionally, synthetic tabular data generation methods may tune the architectural design of the MLP; however, since we are comparing different condensation methods, it is important to use a uniform and consistent architecture. Hence, we adopt an MLP with 2 hidden layers of embedding dimension (100) at a fixed learning rate of 0.001 for a maximum of 100 iterations. Further details on the MLP structure, optimizer, loss, and hyperparameters can be found with the default implementation of Scikit Learn (SKLearn) (https://scikit-learn.org/stable/modules/generated/sklearn.neural_network.MLPClassifier.html)

### A.2.3 DETAILS ON DATASETS

In this work, we benchmarked our methods on 4 tabular datasets from the UCI Machine Learning Repository (https://archive.ics.uci.edu/datasets). In particular, we used: *Adult*, *Shoppers*, *Default*, and *Magic*, all of which are considered classification tasks. Following the work of Zhang et al. (2023), we include their table below (Table 5), describing the statistics of each dataset:

Table 5: Dataset Statistics for the 4 benchmarked UCI tabular datasets. # Num indicates the number of numerical/continuous columns while # Cat indicates the number of categorical columns in the particular dataset..

| Dataset | # Rows | # Num | # Cat | # Train | # Test | Task |
|---|---|---|---|---|---|---|
| **Adult** | $48,842$ | 6 | 9 | $32,561$ | $16,281$ | Classification |
| **Default** | $30,000$ | 14 | 11 | $27,000$ | $3,000$ | Classification |
| **Shoppers** | $12,330$ | 10 | 8 | $11,097$ | $1,233$ | Classification |
| **Magic** | $19,019$ | 10 | 1 | $17,117$ | $1,902$ | Classification |
| **Covertype** | $581,012$ | 54 | 1 | - | - | Multi-Classification |
| **Fourier** | $60,000$ | 86 | 1 | - | - | Multi-Classification |
| **Wine** | 178 | 13 | 1 | - | - | Multi-Classification |

In all cases, the target feature is included as categorical, as we focus on discrete classification tasks in this work. Since our experiments do not involve validation sweeping (i.e., to be fair to all methods we restrict to the same architectural model), we only create the standard train/test split. For privacy and parameter grid search we follow the standard splits obtained from Zhang et al. (2023).

We additionally include the detailed descriptions of each dataset, directly from Zhang et al. (2023).

- **Adult**[1]: Commonly referred to as the UCI Census dataset, this dataset includes a list of financial and demographic information that is used to determine if a particular person's income is greater than $50,000$, hence a binary classification task.

- **Default**[2]: This dataset contains information from credit card accounts in Taiwan between April and September 2005. Using a multitude of features (demographic, credit, etc.), the objective is to determine if the individual will "default" on the following month's credit payment – a binary classification task.

- **Shoppers**[3]: This shopping dataset contains information from an individual web browsing visit with the goal of determining if the client will end up buying something or not – a binary classification task.

---

[1] https://archive.ics.uci.edu/dataset/2/adult
[2] https://archive.ics.uci.edu/dataset/350/default+of+credit+card+clients
[3] https://archive.ics.uci.edu/dataset/468/online+shoppers+purchasing+intention+dataset

- **Magic**[4]: This is a physics dataset that simulates the registration of high-energy gamma particles on the atmospheric Cherekenov gamma telescope, which is ground-based using an imaging-based technique. The objective is to determine the presence of high-energy particles, hence a binary classification task.

- **Covertype**[5]: This dataset describes forest cover type from cartographic variables only.

- **Fourier**[6]: The Fourrier variant is an adaptation of the MNIST dataset that contains the fourier coefficients of the image pixels in tabular format

- **Wine**[7]: The wine dataset contains a list of the chemical characteristics of 3 different types of wine.

### A.2.4 DETAILS ON PRIVACY

As stated in the main paper, for privacy analysis, we use DCR (distance to closest record). This method was adopted from TabSyn's benchmark Zhang et al. (2023). Hence, we apply the same process of privacy evaluation to our condensed datasets. Following TabSyn Zhang et al. (2023) we evaluate DCR in a synthetic versus holdout setting (https://www.clearbox.ai/blog/2022-06-07-synthetic-data-for-privacy-\preservation-part-2). In the case of core-set selection methods, the condensed data is obtained as direct subsets from the training data, meanwhile, the distillation approaches leverage the training data for learning the synthetic tables. In DCR we compare the condensed data with the holdout sets that are not used in the condensation process.

---

[4]https://archive.ics.uci.edu/dataset/159/magic+gamma+telescope
[5]https://archive.ics.uci.edu/dataset/31/covertype
[6]https://archive.ics.uci.edu/dataset/683/mnist+database+of+handwritten+digits
[7]https://archive.ics.uci.edu/dataset/109/wine

