# OpenReview forum: "Exploring Data Distillation for efficient generation of Tabular Data"
_ICLR.cc/2025/Conference — ICLR 2025 Conference Withdrawn Submission_

### Official Review · Reviewer_Zjdy · 2024-10-30

**Soundness:** 2
**Presentation:** 2
**Contribution:** 2
**Rating:** 3
**Confidence:** 3

**Summary:**

In this paper, the authors propose a data distillation method to produce data for the downstream learning tasks. The authors claim that, compared with synthetic data generators, the proposed method is competitive regarding privacy protection and achieving higher utility for downstream learning task. The main idea is utilizing an auto-encoder to map the mixed-data types into a unified latent embedding space and then conduct dataset condensation in the embedding space.

**Strengths:**

**Strength 1.** The experiments are solid. Various datasets and methods are considered in the experiment part. Additionally, the results seem significant compared with existing methods.

**Strength 2.** Compared with synthetic data generators, the proposed method is computationally efficient.

**Weaknesses:**

**Weakness 1.** I do not agree with the claim that privacy protection of the proposed method is stronger than privacy-preserving synthetic data generators. My reason is that if we use the definition of differential privacy (DP) to evaluate privacy protection, then once the synthetic data generator is trained on a synthetic dataset, no matter how many synthetic samples are generated, the privacy protection remains unchanged due to the immunity to post-processing property of DP. However, for the proposed method, the privacy risk will accumulate under differential privacy protection. Using the proposed method 10 times on the same dataset to generate 10 distilled datasets weakens the privacy protection. Therefore, I believe the authors overclaim the advantage regarding privacy protection.


**Weakness 2.** This paper has some writing issues. The most unacceptable thing in this paper is that there are some \ref appearing as question marks. See Page 6, Lines 317-322. **Second**, in this paper, the authors frequently mention image data and the corresponding techniques, which gives me the impression that they simply transfer techniques from the domain of computer vision to a new scenario involving tabular data. As a result, the key contribution is obscured in the writing. **Third**, the authors did not use mathematical notations properly. For example, Equation (1) in Page 3, $[t_0,\ldots, t_N] \in \mathbb{R}^{N\times D}$ makes no sense at all. From my understanding, each $t_i$ is a **column vector** and $[t_0,\ldots, t_N]$ should be $D \times (N+1)$.


**Weakness 3.** The main contribution is to tackle the difficulty of heterogeneity of tabular data. However, the technique of employing a unified embedding space is common in the literature. Therefore, this paper is more like a experimental work validating the effectiveness of some common techniques in tabular data.

**Weakness 4.** In the domain of machine learning, there usually exists a theoretical understanding behind each significant experimental improvement. This paper highlights significant improvements over existing methods but lacks theoretical support. At least the authors should provide some existing results regarding using gradient matching or distribution matching for data condensation.

**Questions:**

**Q1** It is unclear how to condense the embeddings of raw dataset? It would be better to provide detailed algorithms under either gradient matching and attention matching.


**Q2** What is the structure of proxy network in experiments? Does the size of proxy network affect the experimental performance?

---

### Official Review · Reviewer_Jb6G · 2024-11-01

**Soundness:** 2
**Presentation:** 3
**Contribution:** 2
**Rating:** 3
**Confidence:** 4

**Summary:**

In this paper, the authors studied synthetic data generation for tabular data via data distillation method. They proposed a framework to first encode the tabular data through a pre-train feature encoder, then adapted existing data distillation methods developed for image data to distill data in the encoding space, and finally transform the distilled dataset back to the original space via a decoder. They discussed how to adapt 3 different existing data distillation methods, i.e., DM, AM and GM, for tabular data. The novelty part mainly comes from how to adapt existing data distillation methods for tabular data, and the challenging part of the adaptation is not very clear. They also conduct some numerical experiments to compare the proposed framework with existing methods on 7 datasets.

**Strengths:**

The paper studied a very interesting problem of distilling tabular datasets via existing distillation methods developed for image datasets. Overall, the paper is easy to follow. They also investigate multiple existing distillation methods in the context of tabular dataset.

**Weaknesses:**

1. The pre-processing method the paper used for pre-processing the features, a mix of numerical and categorical features, is just following related work Zhang et al. It is not clear if using one-hot encoding and linear transformation of numerical features is good enough. It will be good to perform more studies on the effect of pre-processing methods, like piecewise linear encoding in Gorishniy et al., 2022, etc.
2. The main novelty comes from adapting existing data distillation methods used for image domain to a tabular data context. And other components in the proposed framework are directly from related work TabSyn. This makes the contribution a bit incremental.
3. There is no formal definition of privacy. How to measure privacy leakage of the distilled dataset? No membership inference attack evaluation is performed to verify the possible privacy leakage from model trained on distilled dataset.
4. Evaluations only run for 7 datasets. It is not clear how the authors selected the datasets from UCI. It will be good to see more results on datasets from OpenML.
5. More ablation studies are needed to evaluate how different selection of hyperparameters may affect the final performance of the proposed framework.

**Reference**:
>[Gorishniy et al., 2022] Yury Gorishniy, Ivan Rubachev, and Artem Babenko. On Embeddings for Numerical Features in Tabular Deep Learning. Advances in Neural Information Processing Systems, 35:24991–25004, December 2022

**Questions:**

1. How would the embedding dimensions $D$ affects the data quality after condensation/distillation?
2. Why MLP is used to replace the original proxy networks used for image domain? Any insights on why MLP works better for tabular data?
3. Can the authors add random sampling as a naïve baseline? For all models, such as XGBoost, Random Forest, etc? From Table 3. It looks like “Random” is a very strong baseline for many datasets, except Adult.

---

### Official Review · Reviewer_ApN9 · 2024-11-02

**Soundness:** 1
**Presentation:** 2
**Contribution:** 2
**Rating:** 3
**Confidence:** 4

**Summary:**

This paper investigates generating synthetic tabular data through data condensation and autoencoders. The paper explores three primary distillation strategies: distribution matching, attention matching, and gradient matching—demonstrating, as well as coreset selection methods. The experiment result shows that data generated from condensation methods can achieve comparable downstream classifier performance while saving training time and resources. The authors also conduct a privacy analysis using distance to closest record to analyze the privateness of the distilled data.

**Strengths:**

- Synthetic data generation is indeed a natural extension of data distillation, and this work makes a timely contribution.
- The paper claims substantial improvements in computational efficiency (5x faster) compared to existing synthetic data generation methods. This efficiency gain could make the proposed approach highly relevant for practical applications where compute resources are limited.

**Weaknesses:**

- The paper shows impressive efficiency gains, but it’s unclear how it performs on extremely large datasets. Could the authors provide further information on how their approach scales in terms of memory and latency?
- The privacy is evaluated through distance-to-closest record (DCR), but this metric may not fully capture potential risks of re-identification in real-world scenarios. Including other privacy metrics or discussing potential vulnerabilities would add depth to the privacy claims.
- The experimental evaluation is limited compared to previous tabular works. It would be beneficial to evaluate the proposed method on a broader range of datasets to demonstrate its generalizability and robustness.

**Questions:**

- How does the choice of hyperparameters in the autoencoder impact the quality of the synthesized dataset? It would be helpful if the authors could provide guidance on the sensitivity of the model to hyperparameter changes.
- The paper shows impressive efficiency gains, but it’s unclear how it performs on extremely large datasets. Could the authors provide further information on how their approach scales in terms of memory and latency?
- What is the significance of saving synthesis time? Is there a use case where the multiple synthetic datasets need to be generated for this to be a concern?

---

### Official Review · Reviewer_rzNE · 2024-11-02

**Soundness:** 3
**Presentation:** 3
**Contribution:** 2
**Rating:** 5
**Confidence:** 4

**Summary:**

This paper tackles the dataset distillation task on tabular data. The main challenge typically stems from the mixed data types contained in different cells, such as continuous and categorical values. To address this, the authors initially pre-train a transformer-based VAE and conduct dataset distillation on the latent space like the latent diffusion model. While some details and trade-off discussion are missing, the experiments verify the effectiveness of the proposed method.

**Strengths:**

1. The task is well-motivated, though the experiments do not fully support the motivation.
2. The method is presented well and easy to understand
3. The experiments verify that the proposed method successfully retains the performance after distillation.
4. The idea of distilling on the latent space is interesting despite the additional computation cost

**Weaknesses:**

1. The proposed method requires a pre-trained transformer auto-encoder, which introduces additional computation overhead compared to vanilla dataset distillation approaches. The authors should discuss the trade-off between computation efficiency and efficacy. At the very least, the authors should mention the detailed architectures of the autoencoder. However, I cannot find the details anywhere in the paper, even in the appendix.
1. Following (1), one of the main applications claimed by the authors is efficient parameter parameter search. However, there is no detailed computation cost analysis. I doubt that pre-training a transformer autoencoder may cause more computation costs than directly performing a search on the original dataset. The analysis in L413 is unclear. What is the x and y axes in Figure 5? What would be the FLOPs required by pre-training, distillation, and parameter search?
1. The so-called "privacy" metric DCR is not well-defined. The main paper does not provide sufficient information, making it difficult to judge. It is also unclear why the authors do not report the distance to the closet sample but a modification.
1. As the authors motivate in L104, tabular datasets are often imbalanced. However, the authors only report accuracy in the experiments. I'd recommend the author report F1 and AUC. Otherwise, it is unclear whether the proposed method overfits some classes.
1. How do the authors conduct core-set distillation on tabular datasets in Figure 4? Since the authors have already trained a VAE, K-means can be an informative baseline.

Overall, the work can be interesting and useful in some applications; however, the current form lacks details and contains several editorial flaws.

Editorial comments:
1. The authors misuse \citet and \citep in the entire paper.
2. Many hyperlinks are broken.
3. Figure 3 is never mentioned in the paper.
4. L209 $S_{enc}$ is not defined in Sec. 3-2. There are several typos like this. The authors are encouraged to carefully proofread the paper.

**Questions:**

1. How do the authors get Table 1?
2. In Table 2, why does GM have different trainable parameters compared to the other matching criterion?
3. How does the proposed method initialize the set $\mathcal{S}$?

---

### Official Review · Reviewer_pDQg · 2024-11-04

**Soundness:** 2
**Presentation:** 2
**Contribution:** 2
**Rating:** 3
**Confidence:** 4

**Summary:**

This paper proposes a data distillation method aimed at efficient table data generation. By using a VAE, the original mixed-type data is mapped to the same embedding space, and the data is compressed through a condensation module. This approach reconstructs a smaller dataset that retains only essential features. The effectiveness of the method was experimentally validated by assessing whether the generated datasets maintain the distribution and characteristics of the original data across various compression ratios. Results demonstrate that this approach is more efficient and cost-effective than existing table data generation methods.

**Strengths:**

1. Unlike other methods that generate a full dataset, this approach is original in aiming to efficiently generate data through distillation.
2. The paper proposes three distinct distillation methods, each achieving strong performance, which demonstrates the high quality of the approach.
3. Given its data storage efficiency and cost-effectiveness, this research appears significant.

**Weaknesses:**

1. The numbering of figures and tables is often incorrect or missing, making the paper difficult to follow. Additionally, some sentences in the appendix are incomplete, suggesting the paper is not fully polished.
2. The captions lack sufficient detail for the reader to understand the figures and tables. For instance, the compression ratios (5% and 10%) in Table 1 are not explained.
3. Although the preliminary section claims that the proposed method considers class imbalance, there are no experimental results to support this. Evidence, such as visualizations or results, is lacking to substantiate this claim.
4. There is confusion around the term “DPC.” It appears in sections 2, 4.1, and Figure 4, but does not seem to be used consistently. A clear explanation of this term would have been helpful.

**Questions:**

1. Existing tabular data generation models are not designed to compress the original data. Recent studies [1], [2] show that these models generate data equivalent in size to the original data for evaluation. Considering this, I am curious about the experimental results when the baseline models generate a large volume of data, as in their original studies.
2. In section 3.4, trajectory matching and feature matching are compared. I am curious if all three proposed methods are classified as feature matching. A more detailed explanation of the distillation strategy would have been appreciated.

[1] STaSy: Score-based Tabular data Synthesis, ICLR 2023

[2] TabDDPM: Modelling Tabular Data with Diffusion Models, ICML 2023

---

### Official Review · Reviewer_wGgR · 2024-11-08

**Soundness:** 3
**Presentation:** 1
**Contribution:** 3
**Rating:** 3
**Confidence:** 3

**Summary:**

The authors present a novel synthetic dataset generation pipeline for tabular data that utilizes dataset distillation. Their approach levels a tokenization strategy that would work for both continuous as well as categorical labels, followed by representation learning using a VAE. The authors then use the sample from the latent space of the VAE using several dataset distillation strategies, followed by transformation to the original tabular space (using the VAE decoder). The authors test this approach against both generative models as well as common (non-synthetic) dataset distillation strategies.

**Strengths:**

1) The paper seems methodologically quite novel
2) In the experiments, the proposed pipeline seems to consistently outperform baselines.

**Weaknesses:**

1) The quality of writing needs a lot of improvement. The paper was very difficult to follow at many places and there were trivial mistakes like all the table references appeared as ??. Particularly, the details of how the distillation strategies are adopted to tabular data are quite difficult to understand. Consider re-writing them more clearly and with mathematical details.
2) The preliminaries would benefit from a formal mathematical definition of what dataset distillation is.
3) Please report all experiments need to be repeated multiple (~10) times and report standard errors.
4) Given the authors claims about their method being private, it would be important to experimentally compare privacy loss against generative models.

**Questions:**

See above.

---

### Note · Authors · 2025-01-11

I have read and agree with the venue's withdrawal policy on behalf of myself and my co-authors.